# Probable Dataset Searching Method with Uncertain Dataset Information in Adjusting Architecture Hyper Parameter

## Abstract

Different types of tasks with uncertain dataset information are studied because different parts of data may have different difficulties to achieve. For example, in unsupervised learning and domain adaptation, datasets are provided without label information because of the cost of human annotation. In deep learning, adjusting architecture hyper parameters is important for the model performance and is also time consuming, so we try to adjust hyper parameters in two types of uncertain dataset information:1, dataset labels are postponed to be obtained so hyper parameters need to be adjusted without complete dataset information. 2, hyper parameters are adjusted with a subset training dataset since training models with complete training dataset is time consuming. Here, we propose several loss functions to search for probable dataset when the complete dataset information is not obtained. The experiments on 9 real world data demonstrate the performance of our method.

## 1 Introduction

In deep learning, most regression data can be represented with the form $(\boldsymbol{X}, \boldsymbol{Y})$, where $\boldsymbol{X}$ is the data input and $\boldsymbol{Y}$ is the label. However, different parts of the data may have different difficulties to achieve. For example, in unsupervised learning Barlow (1989) and domain adaptation Wang & Deng (2018), the label of dataset is assumed hard to be obtained since label usually needs human annotation. These situations could be viewed as making decisions when part of dataset information is uncertain. Another probable situation is that input sample $\boldsymbol{X}$ can be obtained much earlier than label $\boldsymbol{Y}$ is obtained, because human annotation of $\boldsymbol{Y}$ is time consuming or the exact task target of $\boldsymbol{Y}$ is not determined when the input samples are collected. In such situation, the computing resources are assumed to be abundant before the label $\boldsymbol{Y}$ is obtained. In deep learning, architecture hyper parameter setting is an important factor for the performance of a model. Then a question is whether the architecture hyper parameters, corresponding to different network architectures, can be compared and adjusted only with input sample information $\boldsymbol{X}$. Selecting hyper parameter only with input sample information $\boldsymbol{X}$ could save the time to try different hyper parameters when the label $\boldsymbol{Y}$ is obtained. Input sample information $\boldsymbol{X}$ usually takes more memory space than label information $\boldsymbol{Y}$, indicating that input sample information $\boldsymbol{X}$ contains more information than label information $\boldsymbol{Y}$, so predicting the architecture comparison only with input sample information $\boldsymbol{X}$ seems probable.

To deal with the uncertain information in a dataset, we propose a probable dataset searching method to predict architecture comparison, where the dataset representation is inspired by the dataset definitions and assumptions in recent neural network convergence works Kohler & Langer (2021); Bauer & Kohler (2019); Schmidt-Hieber (2020); Suzuki (2018); Farrell et al. (2021). Our method could search probable datasets with provided dataset information such as input sample information $\boldsymbol{X}$. Concretely, the comparison of two hyper parameters can be predicted by searching for the existence of probable dataset that one architecture is better or worse than another. Here, we use a neural network to approximate the dataset regression function and apply several loss functions to search for the probable dataset that a trained architecture is better than another in testing dataset.

An assumption in our method is that the compared architectures should have competitive performance on searched dataset. Empirically, the compared architectures are selected because they perform well

on the data domain due to former experience, so we search the dataset from the situations that at least one compared architecture could perform well in the dataset. In implementation, the neural network to approximate the regression function has the same architecture as a compared architecture.

Probable dataset searching method could also help analyze the characteristics irrelevant to the concrete dataset information of $X$. For example, sometimes training models with complete dataset costs too much time so a small subset can be used to approximately adjust architecture hyper parameters Klein et al. (2017); Elsken et al. (2019). However, this approximation is correct sometimes and wrong at other times. When comparing two given hyper parameters, probable dataset searching method could figure out the conditions that the approximate comparison is correct.

## 2 METHOD

### 2.1 DATASET REPRESENTATION

When a sample in a regression dataset is denoted by a random variable vector with the form $(X, Y)$, the relationship between $X$ and $Y$ can be represented as a regression function $f_0(x) = \mathbf{E}\{Y|X = x\}$, where $\mathbf{E}$ is the expectation. In recent deep neural network convergence theorem Kohler & Langer (2021); Bauer & Kohler (2019); Schmidt-Hieber (2020); Suzuki (2018); Farrell et al. (2021), it is proved that $f_0$ could be converged by multi-layer fully connected neural networks trained with enough samples, where $f_0$ satisfies some requirements and assumptions. Here, we assume that the convergence theorem is also true with other types of deep neural networks. We then use a neural network to approximate a regression function with two reasons: 1, as a function, neural network also satisfies the requirements and assumptions of a regression function. 2, there is always a neural network $g$ that the difference between $f_0$ and $g$ is smaller than a given positive value because of the definition of convergence.

As a result, a regression dataset can be expressed by
$$Y = g(X) + \epsilon, \tag{1}$$
where the neural network $g$ has a small enough similarity to $f_0$, $\epsilon$ is a random disturbance of regression function with 0 expectation. Then, there are three components influencing the regression dataset, the regression function $g$, $X$ distribution and random disturbance distribution $\epsilon$. $\epsilon$ should be much smaller than $g$ in the dataset so the relation between inputs and outputs could be easily distinguished from random disturbance.

There is another additional assumption of the regression dataset: The regression function should not be too complex that no compared architecture can perform well in the comparison. It also assumes that the compared architectures in the searching method are selected with empirical knowledge to provide competitive performance in an uncertain dataset with enough training samples. Practically, the assumption is satisfied by setting the regression function $g$ with the same architecture as a compared architecture.

Formally, to train a neural network $g_{\text{AR-A}}$ with its parameters $\theta_{\text{AR-A}}$ with a training dataset $\alpha_1$, error value Err is introduced as a smaller the better value to evaluate the performance of a model in the dataset $\alpha_1$. Then training $g_{\text{AR-A}}$ with a training dataset $\alpha_1$ aims to find parameters $\theta^*_{\text{AR-A}}(\alpha_1)$ that
$$\theta^*_{\text{AR-A}}(\alpha_1) = \arg\min_{\theta_{\text{AR-A}}} \text{Err}(\alpha_1, g_{\text{AR-A}}(\cdot, \theta_{\text{AR-A}})), \tag{2}$$
where smaller error value means better model performance in training dataset $\alpha_1$. In regression task, Root Mean Square Error (RMSE) is a widely used error value: $\text{RMSE} = \sqrt{\frac{1}{K}\|\hat{y} - y\|_2^2}$, where $\|\cdot\|_2$ is the $l$-2 norm, $y$ is the true value, $\hat{y}$ is the model output and $K$ is the dimension of $y$.

The difference of the error value between two architectures on a testing dataset $\alpha_2$ can be used to compare the performance of two architectures AR-A and AR-B:
$$C(\text{AR-A}, \text{AR-B}, \alpha_1, \alpha_2) = \text{Err}(\alpha_2, g_{\text{AR-A}}(\cdot, \theta^*_{\text{AR-A}}(\alpha_1))) - \text{Err}(\alpha_2, g_{\text{AR-B}}(\cdot, \theta^*_{\text{AR-B}}(\alpha_1))), \tag{3}$$
where models are trained by training dataset $\alpha_1$ and tested by testing dataset $\alpha_2$.

In this paper, we try to compare the performance of two architectures when part of dataset information is uncertain, which aims to find the probable situations that AR-A (or AR-B) performs better, corresponding to a small $C(\text{AR-A}, \text{AR-B}, \alpha_1, \alpha_2)$ (or $C(\text{AR-B}, \text{AR-A}, \alpha_1, \alpha_2)$) value.

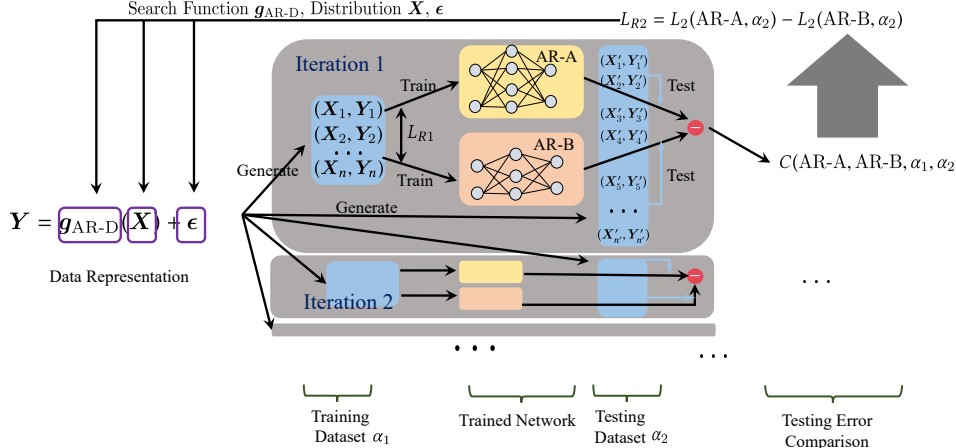

Figure 1: Probable dataset searching procedure, where models are trained in $\alpha_1$ and tested in $\alpha_2$. The testing difference of two architectures are used to search probable dataset.

## 2.2 CALCULATION AND LOSS FUNCTION

We start with the illustration of loss functions, which are important to search for dataset parameters in backpropagation. As shown in Fig. 1, by optimizing loss functions, we could search for the probable dataset that architecture AR-A performs better than AR-B. Then, the comparison score can be used to test the performance of a searched dataset.

In regression task, $L_2$ or mean square error loss is a widely used loss function to train deep learning models, where a small $L_2$ loss means a small error value. The $L_2$ loss function to train an architecture AR-A with dataset $\alpha$ can be denoted by:

$$L_2(\text{AR-A}, \alpha) = \frac{1}{nK} \sum_{i=1}^{n} \|\boldsymbol{g}_{\text{AR-A}}(\boldsymbol{X}_i, \boldsymbol{\theta}_{\text{AR-A}}) - \boldsymbol{Y}_i\|_2^2, \tag{4}$$

where $(\boldsymbol{X}_i, \boldsymbol{Y}_i) \in \alpha$, dataset $\alpha$ contains $n$ samples, $K$ is the dimension of $\boldsymbol{Y}$, $\| \cdot \|_2$ denotes the $l$-2 norm and $\boldsymbol{g}_{\text{AR-A}}$ is a neural network with architecture AR-A and parameters $\boldsymbol{\theta}_{\text{AR-A}}$.

When we deal with the situation that the dataset do not have label $\boldsymbol{Y}$ so the information of regression function is uncertain, we use a neural network with architecture AR-D to approximate the regression function $\boldsymbol{g}$ in Eq. equation 1. Then, when representing the dataset with the form of Eq. equation 1, $L_2$ loss becomes $L_2(\text{AR-A}, \alpha) = \frac{1}{nK} \sum_{i=1}^{n} \|\boldsymbol{g}_{\text{AR-A}}(\boldsymbol{X}_i, \boldsymbol{\theta}_{\text{AR-A}}) - (\boldsymbol{g}_{\text{AR-D}}(\boldsymbol{X}_i, \boldsymbol{\theta}_{\text{AR-D}}) + \boldsymbol{\epsilon}_i)\|_2^2$, where $L_2$ can also be used to train the variables that contain dataset information, including the parameters $\boldsymbol{\theta}_{\text{AR-D}}$, the distribution of $\boldsymbol{X}_i$ and $\boldsymbol{\epsilon}_i$.

To compare the performance, two architectures AR-A and AR-B first need to be trained with the same training dataset $\alpha_1$ with $L_2$ loss:

$$\min L_{R1} = \min_{\boldsymbol{\theta}_{\text{AR-A}}} L_2(\text{AR-A}, \alpha_1) + \min_{\boldsymbol{\theta}_{\text{AR-B}}} L_2(\text{AR-B}, \alpha_1), \tag{5}$$

where the parameters of AR-A and AR-B reach $\boldsymbol{\theta}_{\text{AR-A}}^*$ and $\boldsymbol{\theta}_{\text{AR-B}}^*$ with minimum $L_{R1}$ value.

If we want to find a probable dataset where AR-A is better than AR-B, we need to let the comparison score $C(\text{AR-A}, \text{AR-B}, \alpha_1, \alpha_2)$ small. Since small $L_2$ loss means small error value, searching a dataset for a small comparison score equals to search $\boldsymbol{\theta}_{\text{AR-D}}$, $\boldsymbol{X}$ and $\boldsymbol{\epsilon}$ with loss

$$\min L_{R2}(\text{AR-A}, \text{AR-B}) = \min_{\boldsymbol{\theta}_{\text{AR-D}}, \boldsymbol{X}, \boldsymbol{\epsilon}} (L_2(\text{AR-A}, \alpha_2) - L_2(\text{AR-B}, \alpha_2)), \tag{6}$$

where the parameters of AR-A and AR-B are trained by $L_{R1}$ and not influenced by $L_{R2}$. As illustrated in the additional assumption, we let the dataset regression function architecture AR-D be the same with AR-A in $L_{R2}$ since we want to search for the dataset that AR-A performs well and a dataset with architecture AR-A should best fit the AR-A model.

When dataset information needs to be searched, both $\alpha_1$ and $\alpha_2$ obey the relation described in Eq. equation 1, so the dataset searched with $L_{R2}$ loss function also influence $\alpha_1$ in $L_{R1}$ loss. Although $L_2$ is differentiable with the respect to the parameters related to probable dataset, the $L_{R1}$ loss only influence the parameters in AR-A and AR-B when training models. Besides, no parameters in AR-A and AR-B are influenced by the $L_{R2}$ loss when searching dataset. The combination of $L_{R1}$ and $L_{R2}$ losses means searching the dataset that a trained AR-A model performs better than a trained AR-B model in testing dataset. We discuss more details of optimizing $L_{R2}$ in Appendix.

$X$ distribution, regression function $g_{\text{AR-D}}$ and $\epsilon$ are three unknown variables in Eq. equation 1 that can be searched in our method. In this paper, the dataset searching method can be applied to different types of dataset information missing. For example, when the dataset only does not have label $Y$, the $X_i$ in loss functions uses the known dataset input information and only parameters $\theta_{\text{AR-D}}$ and $\epsilon$ needs to be searched in Eq. equation 6. If the dataset does not have label $Y$ and complete $X$ distribution, $X$ distribution also needs to be searched in Eq. equation 6. Concretely, the parameters $\theta_{\text{AR-D}}$ in regression function $g_{\text{AR-D}}$ can be trained by backpropagation algorithm since $L_{R2}$ is differentiable with the respect to $\theta_{\text{AR-D}}$ in $\alpha_2$. If $X$ distribution needs to be searched, we use a set of discrete input samples $\beta_X$ as an approximation to represent the distribution of $X$, where the elements of $\beta_X$ obey the distribution of $X$. The size of set $\beta_X$ should be a large number (100,000 in our experiment) for a fine approximation. Then input samples $X_i$ in datasets $\alpha_1, \alpha_2$ can be generated by randomly selecting inputs $X$ from $\beta_X$ and corresponding $Y$ can be calculated with Eq. equation 1, where the size of $\alpha_1$ is set to a fixed number to discover the relationship between train size and architecture difference. The distribution of $X$ can be searched by adjusting the input samples $X_i$ in set $\beta_X$ through the backpropagation of input samples $X_i$ in dataset $\alpha_2$ with $L_{R2}$ loss, where $\alpha_2$ is generated from the set $\beta_X$ repeatedly. Similarly, the distribution of $\epsilon$ can be trained by adjusting the value in a finite set as an approximation.

When training the distribution of $X$ in $\beta_X$, we need an additional normalization procedure since current calculation environment can not deal with number with large out of range value (for example, single or double precision floating point format both have maximum value). As a result, when searching $X$ distribution, we apply a normalization method, which let each dimension in $X$ has $0$ mean and $1$ standard deviation. The normalization equals to a linear operation to each dimension in $X$. For distribution of $\epsilon$, we normalize $\epsilon$ with $0$ mean and a small standard deviation since the expectation of $\epsilon$ should be $0$ and we aim to study the situation that the regression function has the main influence on the value of $Y$.

Similarly, the searched output value of regression function should not out of range, so we normalize the searched regression function $g_{\text{AR-D}}$ with loss

$$L_{RN} = \|\text{mean(AR-D}, \alpha_2) - \mathbf{0}\|_2^2 + \|\text{std(AR-D}, \alpha_2) - \mathbf{1}\|_2^2, \tag{7}$$

where $\text{mean(AR-D}, \alpha_2)$ (or $\text{std(AR-D}, \alpha_2)$) means calculating the mean (or standard deviation) vector of each dimension in $g_{\text{AR-D}}$ on all input samples $X_i$ in dataset $\alpha_2$. $L_{RN}$ aims to normalize the output of regression function with $0$ mean and $1$ standard deviation to avoid out of range large number in searching process.

In implementation, we search dataset through loss $L_{R1}$, $L_{R2}$ and $L_{RN}$ with several iterations, where in each iteration, the parameters in architectures AR-A and AR-B are randomly initialized and trained. The searched dataset will be tested by the score $C(\text{AR-A}, \text{AR-B}, \alpha_1, \alpha_2)$ for evaluation.

## 3 EXPERIMENTS

### 3.1 ADJUSTING HYPER PARAMETER WITH INPUT INFORMATION

Here, we want to evaluate several methods in comparing two architecture hyper parameters without knowing complete data information. These methods aim to save the time of adjusting hyper parameters after obtaining the complete data information. When comparing two hyper parameters denoted by architectures AR-A and AR-B with uncertain dataset information, different comparison predictions lead to different training preferences after complete data information is obtained. There are three types of preferences: 1, only training AR-A with complete data. 2, only training AR-B. 3, training both and selecting the better. Three actually training preferences correspond to different comparison prediction explanations: Corresponding to type 1 (or 2) preference, architecture AR-A (or AR-B) is

always better than or similar to another in probable datasets, so only one architecture needs to be actually trained. Corresponding to type 3 preference, architecture AR-A is sometimes better and sometimes worse than AR-B in probable datasets. Type 1 and 2 comparisons adjust and select hyper parameters before the complete data information is obtained while type 3 comparison is the most conservative prediction that adjusts hyper parameters after obtaining the complete data information.

There are two metrics in evaluating the performance of a comparison prediction, correct (or wrong) situation and actually training times. Here, if an architecture AR-A actually performs better than another in a complete dataset and AR-A will (or will not) be actually trained in a comparison prediction, the comparison prediction is correct (or wrong). If an architecture AR-A actually performs similar with another, any actually trained preference is acceptable so any comparison prediction is correct. Here, if two architectures do not have statistical significance in a dataset with complete information, we treat them as similar architectures, where we apply a $5\%$ significance level $t$-test score with 10 experiments for significance test.

When considering two metrics, correct prediction times is larger the better and actually training times is smaller the better since we want to train models with smaller time consumption of adjusting hyper parameters after obtaining complete dataset information. We apply an Average Evaluation Score (AES) to consider two metrics together when making multiple comparison predictions:

$$\text{AES} = \frac{1}{N}(\text{Correct} - 2\text{Wrong} - 0.5\text{Training}), \tag{8}$$

where $N$ is comparison predictions number. Correct (or Wrong) refers to the correct (or wrong) prediction times. Training refers to the total actually training times in the comparison predictions. In AES, the weight of training times is half of the correct prediction times to let the most conservative type 3 prediction has 0 evaluation score. The weight of wrong prediction times is twice of the correct prediction times since a wrong comparison prediction means selecting a significantly worse architecture. Then, it needs more time and computing resources to discover and rectify a wrong prediction.

### 3.1.1 DATASETS AND HYPER PARAMETER CANDIDATES

We test the performance of different comparison prediction methods on time series forecasting datasets, where the time series value of a future time step is predicted by a multidimensional time series sequence. In time series forecasting, time step values of different time intervals can be predicted by the same sequence, for example, a sequence from time step 1 to 100 can be used to predict the value of time step 101, 102, 103. Then, using the series to predict the value of different time intervals are the datasets with the same input $X$ and different label $Y$.

Then, the experiments aim to predict the selection of different Long Short Term Memory (LSTM) hidden dimension numbers, where LSTM Hochreiter & Schmidhuber (1997) is a commonly used architecture for time series forecasting datasets and the hidden dimension number is a commonly adjusted hyper parameters in LSTM network. Probably the best hidden dimension numbers are related to the input dimension, so we study the series with a fixed 6 dimension and 100 series length. The candidate hidden dimension numbers are 4, 32, 256, which correspond to the hidden dimension similar to the time series dimension, the hidden dimension a few times larger than the time series dimension, the hidden dimension much larger than the time series dimension. We use the last hidden state of LSTM to predict regression output by a 2-layer fully connected neural network with 50 hidden units and ReLU activation function. There are two comparison prediction tasks, where the first makes hyper parameter selection between hidden dimension number 32 and 256, the second makes selection between dimension number 4 and 32.

The experiments are made on nine different time series data with three data domains. Air: Three air quality data contain time series of daily air quality indexes collected from 1 Jan. 2014 to 1 Mar. 2022 in the Shanghai, Beijing and Shenzhen cities of China[1]. WormMotion: Three organism motion data correspond to three mutant types (wild, goa-1, unc-38) of time series extracted from EigenWorms dataset in UEA time series classification archive Bagnall et al. (2018). HandMEG: Three brain activity data contain the Magnetoencephalography (MEG) time series with three types of hand and wrist movement (left, right, up) extracted from HandMovementDirection dataset in UEA time series

---

[1] https://aqicn.org/data-platform/

classification archive, where we select first 6 dimensions from 10 for forecasting to since we want to keep the ratio between data dimension number and candidate LSTM hidden dimension numbers consistent in different data. For each data, forecasting the time series with 1,2,3 and 4 intervals corresponds to four regression datasets with the same $X$ and different $Y$. More training samples usually has positive influence on model performance, but whether larger training dataset size could provide more concise architecture comparison prediction is uncertain. So we make experiments on various train size, from 200, 400, 800 to 1600. The validation and testing dataset size are 500. As a result, there are total 144 datasets in the experiments.

### 3.1.2 BASELINES

• CP: Conservative Prediction (CP) does not make any preference between two architectures, so conservative prediction always make type 3 prediction. CP does not require any dataset information.

• SDP 1: Similar Distribution Prediction 1 (SDP 1) comes from the simple idea that the hyper parameter performs well in a dataset will also performs well in a similar dataset. Here, we apply SDP 1 to predict the hyper parameter of a dataset named dataset 2 based on another dataset 1, where two datasets have exactly the same $X$ and different label $Y$ (here dataset 1 and 2 are datasets with different forecasting intervals extracted by the same time series data). SDP 1 predicts hyper parameter of dataset 2 based on statistical significance on dataset 1, which makes type 1 (or type 2) prediction on dataset 2 if a hyper parameter named AR-A performs significantly better (or worse) on data 1. If two hyper parameters do not have significantly difference on dataset 1, SDP 1 will make type 3 prediction on data 2.

• SDP 2: Similar Distribution Prediction 2 (SDP 2) needs the same dataset information as SDP 1 and makes bold prediction based on error difference on data 1. SDP 2 makes type 1 (or type 2) prediction on dataset 2 if AR-A performs better (or worse) on data 1, no matter whether the performance is significant or not.

In our method, architecture comparison prediction can be made by searching probable datasets with loss in Eq. equation 6. When comparing the architecture hyper parameters denoted by AR-A and AR-B, two types of probable datasets need to be searched, including the datasets that AR-A performs better and the datasets that AR-B performs better. Then the comparison prediction is based on the searching result: If a searched probable dataset shows statistically significantly better performance of AR-A in testing dataset (after trained and validated in training and validation dataset) and no searched probable dataset shows statistically significantly better performance of AR-B, the method makes type 1 prediction. Similarly, if a probable dataset that AR-B is significantly better exists and no probable dataset that AR-A is significantly better exists, the method makes type 2 prediction. If the dataset that AR-A is significantly better and the dataset that AR-B is significantly better both can be searched, the method makes type 3 prediction. Here, we apply the probable dataset searching method to predict architecture comparison with different levels of uncertain dataset information.

• PDS 0: Probable Dataset Searching 0 (PDS 0) method searches dataset without any dataset information, so all parts of dataset information need to be searched in PDS 0.

• PDS 1: Probable Dataset Searching 1 (PDS 1) method searches dataset without any information of label $Y$, so the regression function needs to be searched in the situation. Besides, PDS 1 has the input samples $X$ information in the dataset but the training/validation/testing split of input samples $X$ is unknown and shuffled in PDS 1. In time series forecasting task, the true training/validation/testing split of input samples $X$ is an important information since the input samples in each split are actually similar: In time series forecasting, input samples are extracted by a sliding window and adjacent input samples are usually put in the same split since training set should not contain the forecasting target information of validation and testing set. Then the situation that input samples in each training/validation/testing set are similar contains more information than the situation that samples in each training/validation/testing set are shuffled.

• PDS 2: Probable Dataset Searching 2 (PDS 2) method searches dataset regression function with the input samples $X$ information, including training/validation/testing split. Similar to PDS 1, the method does not require any information of label $Y$.

Here, different methods require different levels of dataset information, which can be ordered as follows: CP = PDS 0 < PDS 1 < PDS 2 < SDP 1 = SDP 2. SDP 1 and SDP 2 require most detailed

Table 1: Comparison prediction AES of several methods.

| Comparison Task | Dataset Domains | Train Size | CP | SDP 1 | SDP 2 | PDS 0 | PDS 1 | PDS 2 |
|---|---|---|---|---|---|---|---|---|
| 32 Dim vs 256 Dim | Air | 200 | 0 | 0.083 | 0.167 | 0 | 0 | **0.5** |
| | | 400 | 0 | 0.25 | 0.167 | 0 | 0 | **0.5** |
| | | 800 | 0 | 0.25 | 0.333 | 0 | 0 | **0.5** |
| | | 1600 | 0 | 0.333 | **0.5** | 0 | 0 | **0.5** |
| | WormMotion | 200 | 0 | 0.417 | **0.5** | 0 | 0 | 0.333 |
| | | 400 | 0 | **0.375** | 0.333 | 0 | 0 | 0.333 |
| | | 800 | 0 | 0.292 | **0.333** | 0 | 0 | **0.333** |
| | | 1600 | 0 | 0.208 | **0.417** | 0 | 0 | 0.333 |
| | HandMEG | 200 | **0** | -0.125 | -0.333 | **0** | **0** | **0** |
| | | 400 | **0** | -0.5 | -0.75 | **0** | **0** | **0** |
| | | 800 | **0** | -0.5 | -0.833 | **0** | **0** | **0** |
| | | 1600 | 0 | -0.125 | -0.167 | 0 | 0 | **0.167** |
| 4 Dim vs 32 Dim | Air | 200 | 0 | 0.083 | 0.25 | 0 | **0.5** | **0.5** |
| | | 400 | 0 | 0.083 | 0.333 | 0 | **0.5** | **0.5** |
| | | 800 | 0 | 0.083 | 0.417 | 0 | **0.5** | **0.5** |
| | | 1600 | 0 | 0.125 | 0.167 | 0 | **0.5** | **0.5** |
| | WormMotion | 200 | 0 | 0.333 | 0.333 | 0 | **0.5** | **0.5** |
| | | 400 | 0 | 0.375 | 0.333 | 0 | **0.5** | **0.5** |
| | | 800 | 0 | **0.5** | **0.5** | 0 | **0.5** | **0.5** |
| | | 1600 | 0 | **0.5** | **0.5** | 0 | **0.5** | **0.5** |
| | HandMEG | 200 | 0 | 0.208 | -0.167 | 0 | 0 | **0.5** |
| | | 400 | 0 | 0.25 | -0.25 | 0 | 0 | **0.5** |
| | | 800 | 0 | 0.458 | **0.5** | 0 | 0 | **0.5** |
| | | 1600 | 0 | **0.5** | **0.5** | 0 | 0 | **0.5** |
| | Total AES | | 0 | 0.186 | 0.17 | 0 | 0.167 | **0.396** |
| | Avg. Correct | | 1 | 0.968 | 0.89 | 1 | 1 | 1 |
| | Avg. Wrong | | 0 | 0.032 | 0.11 | 0 | 0 | 0 |
| | Avg. Training Times | | 2 | 1.434 | 1 | 2 | 1.667 | 1.208 |

dataset information including the input samples $X$ information and the label $Y$ information on a similar dataset. Probable dataset searching methods do not require such label $Y$ information since that information usually can not be easily obtained in real world situation. Besides, CP and PDS 0 requires no specific information of the dataset.

### 3.1.3 RESULTS

Table 1 shows the AES results of three data domains on two hyper parameter comparison tasks, with the summarized information listed in the bottom 4 lines, including average AES, average correct times, average wrong times and average actually training times. The AES result of each data domain is averaged by the AES of three data in the domain, and each data corresponds to four datasets with different forecasting intervals. For each dataset, there are other three datasets share the same input samples $X$ and have different $Y$, so SDP 1 and 2 methods could make three comparison predictions with each dataset. The best AES score is highlighted in bold.

Firstly, it can be found that the probable dataset searching method 2 (PDS 2) achieves the best AES score while it requires less dataset information than baselines SDP 1 and 2. The predictions made by PDS 2 could make high level of correct prediction times with small actually training times.

Secondly, the performance of PDS 0 is the same with the performance of CP, because when no dataset information is provided, each candidate hyper parameter may perform better due to the datasets searched by the method. As a result, in the experiments, PDS 0 always makes type 3 prediction for each dataset as the CP method does.

Thirdly, the difference of AES score among PDS 0, PDS 1 and PDS 2 shows that with more dataset information, the probable dataset searching method could provide better comparison prediction between hyper parameters.

Fourthly, SDP 1 and 2 perform better than CP, PDS 0 and PDS 1 since they consider more dataset information. However, SDP 1 and 2 are not practical methods because a similar dataset with the same input samples $X$ information is hard to obtain. Besides, the predictions made by SDP 2 decrease the actually training times but increase the wrong prediction times comparing to SDP 1. Since a wrong

Table 2: Comparison prediction by a small training subset to select hidden dimension between 32 and 256, where 32 (or 256) ↑ means 32 (or 256) performs better in RMSE error value. Sig. (or No Sig.) means the difference is statistically significant (or not). The underlined number means the corresponding situation could be searched by PDS, for example, 6 in row 32 ↑ & Sig. means PDS method could find the dataset where 32 dimension is significantly better in both 200 and 1600 train size.

| | | Situation Happened Times | | | Prediction Methods | | | |
| | | Train Size 1600 | | | Correct/Training Times/AES | | | |
| | | 32 ↑ & Sig. | 256 ↑ & Sig. | No Sig. | CP | SDP1 | SDP2 | PDS |
|---|---|---|---|---|---|---|---|---|
| | 32 ↑ & Sig. | 6 | 2 | 3 | 1/22/0 | 0.818/11/-0.045 | 0.818/11/-0.045 | 1/22/0 |
| Train Size | 256 ↑ & Sig. | 0 | 0 | 5 | 1/10/0 | 1/5/0.5 | 1/5/0.5 | 1/5/0.5 |
| 200 | 32 ↑ & No Sig. | 4 | 1 | 6 | 1/22/0 | 1/22/0 | 0.909/11/0.227 | 1/22/0 |
| | 256 ↑ & No Sig. | 4 | 1 | 4 | 1/18/0 | 1/18/0 | 0.556/9/-0.833 | 1/18/0 |
| | Total AES | | | | 0 | 0.056 | -0.083 | **0.069** |

prediction could lead to severe consequence in hyper parameter selection, SDP 2 performs worse than SDP 1 in AES score.

Finally, increasing the training dataset size could not apparently improve the hyper parameter comparison prediction performance, although larger training dataset size means more information for SDP 1, SDP 2, PDS 1 and PDS 2 methods. We provide some additional illustration in Appendix.

## 3.2 ADJUSTING HYPER PARAMETER WITH A TRAINING SUBSET

Another situation of selecting hyper parameters without complete dataset is using a subset of training dataset to adjust hyper parameter. Sometimes the training dataset is large so using a subset to adjust hyper parameters is an efficient approximation. Since the approximation may be wrong sometimes, we apply the probable dataset searching method to check the conditions when the approximate comparison is correct. The probable dataset searching method in this part does not require any information of input samples $X$ so the procedure to find the conditions do not influence the efficiency of the approximate comparison. As a result, probable dataset searching method here is actually PDS 0, where all parts of dataset information need to be searched.

In this part, we apply the probable dataset searching method to find whether two architectures have different performances in different train sizes. Denote the training sets in the same dataset with two train sizes by $\alpha_1$ and $\alpha_1'$. Then, training architectures in different training sets correspond to optimize the loss $L_{ST} = L_{R1}(\text{AR-A}, \text{AR-B}, \alpha_1) + L_{R1}(\text{AR-A'}, \text{AR-B'}, \alpha_1')$, where AR-A/AR-A' (AR-B/AR-B') are neural networks with the same architecture and different parameters $\theta$. AR-A/AR-A'/AR-B/AR-B' are tested in the same testing set $\alpha_2$.

To search for different probable situations of architecture comparisons in two train sizes, we needs different kinds of loss functions. For example, to optimize a situation that AR-A performs better than AR-B in training set $\alpha_1$ while they performs similar in training set $\alpha_1'$, the loss function is $L_{S1} = L_{R2}(\text{AR-A}, \text{AR-B}) + \frac{1}{nK} \sum_{i=1}^{n} \|g_{\text{AR-A'}}(X_i, \theta_{\text{AR-A'}}) - g_{\text{AR-B'}}(X_i, \theta_{\text{AR-B'}})\|_2^2$, where $X_i$ is the input sample in $\alpha_2$, $n$ is the total test size, $K$ is the dimension of $g$.

Another probable situation is AR-A performs better than AR-B in training set $\alpha_1$ while AR-B performs better in training set $\alpha_1'$, which corresponds to the loss $L_{S2} = L_{R2}(\text{AR-A}, \text{AR-B}) + L_{R2}(\text{AR-B'}, \text{AR-A'})$. For the situation that AR-A performs better than AR-B in training set $\alpha_1$ and $\alpha_1'$, the loss is $L_{S3} = L_{R2}(\text{AR-A}, \text{AR-B}) + L_{R2}(\text{AR-A'}, \text{AR-B'})$.

### 3.2.1 RESULTS

Table 2 shows the experimental results, where we use 1600 and 200 train size pairs of all the datasets in the former experiment, which contains totally 36 dataset pairs. In this part, different methods use the training dataset with 200 size to predict the hyper parameter selection in train size 1600. We list the happened times of different situations in dataset pairs in the left part of the table, for example, 6 dataset pairs satisfy the situation that 32 hidden dimension is significantly better than 256 hidden dimension with both 200 and 1600 train size. The performance of four methods in different situations

is listed in the right part of the table, for example, for the 11 dataset pairs that 32 hidden dimension is significantly better with 200 train size, the average AES of SDP 1 is -0.045.

Here, similar to the former experiments, there are also three types of comparison predictions. Conservative Prediction (CP) always makes type 3 prediction. SDP 1 predicts comparison based on statistical significance with train size 200, where SDP 1 make type 3 prediction if no significantly difference is observed with train size 200. SDP 2 predicts comparison based on error difference with train size 200. PDS makes prediction based on the architecture statistical difference with train size 200 and the searched probable situations with train size 1600. For example, when 32 hidden dimension is significantly better with 200 train size, both 32 and 256 hidden dimension may be significantly better in PDS (6 and 2 are underlined in the table), so PDS makes type 3 prediction.

The result shows that PDS performs best among baselines including CP, SDP 1 and SDP 2. In most situations, SDP 1 performs similar to PDS, which means SDP 1 is still a good approximation comparison method. SDP 1 method performs worse than PDS since it makes more wrong predictions when 32 hidden dimension is significantly better with 200 train size, which also shows that selecting hyper parameter from a subset training dataset is not always correct.

## 4 RELATED WORKS

**Uncertain Dataset Information.** In application, different parts of dataset information have different difficulties to obtain, which leads to several types of task with uncertain dataset information. In unsupervised learning Barlow (1989); Creswell et al. (2021); Baevski et al. (2021); Yu et al. (2021); Choudhury et al. (2021), data labels are assumed hard to be obtained so models are expected to work without data labels. In domain adaptation Wang & Deng (2018); Dong et al. (2021); Zellinger et al. (2021); Chen & Chao (2021); Lv et al. (2021); Rostami (2021), complete dataset information in source data domain and unlabeled dataset in target data domain are used to train models in target data domain. In few-shot learning Wang et al. (2020); Snell et al. (2017); Wang et al. (2021); Sendera et al. (2021); Cao et al. (2021), few train size dataset is used to train models with additional information of a similar dataset. In this paper, we discuss the adjustment of architecture hyper parameters with two kinds of uncertain dataset information: 1, dataset labels are postponed be obtained so hyper parameters need to be adjusted without complete dataset information. 2, adjusting hyper parameters with a subset training dataset since training models with complete training dataset is time consuming.

**Regression Dataset Representation.** Recent works Kohler & Langer (2021); Bauer & Kohler (2019); Schmidt-Hieber (2020); Suzuki (2018); Farrell et al. (2021) show that if a regression function $f_0$ satisfies some requirements, it can be converged by a multi-layer fully connected neural network. Although the requirements are different in different works (such as Hölder function requirements Schmidt-Hieber (2020) and $(p,C)$-smooth requirements Bauer & Kohler (2019)), they include a wide range of function types so these requirements are assumed to be satisfied by the regression function in real world functions. Our searching method use a neural network to approximate function $f_0$ and assume that the convergence is also true in other types of neural network. Our regression dataset representation further assumes that at least a compared architecture has competitive performance when the compared architectures worth for application.

## 5 CONCLUSION

In this paper, we discuss adjusting hyper parameters in two types of uncertain dataset information, including the postponed dataset labels and only using a subset training dataset. To infer the probable complete dataset information, we propose a probable dataset searching method which searches the variables in our dataset representation. We test the performance of our method in 9 real world data with two types of uncertain dataset information.

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

# A    APPENDIX

## A.1    DATASET INFORMATION SEARCHING DETAILS

In Eq. (6), the searched variables $\boldsymbol{\theta}_{\text{AR-D}}$, $\boldsymbol{X}$ and $\boldsymbol{\epsilon}$ influence $L_{R2}$ in two aspects: 1, they directly influence testing dataset $\alpha_2$ in Eq. (6). 2, they indirectly influence training dataset $\alpha_1$, which influence the parameters $\boldsymbol{\theta}_{\text{AR-A}}$ and $\boldsymbol{\theta}_{\text{AR-B}}$ in Eq. (6).

The parameters of AR-A and AR-B are trained by $\alpha_1$ in $L_{R1}$ and these parameters influence the loss $L_{R2}$. When parameters are searched by gradient decent, each step of training the $i$-th element of $\boldsymbol{\theta}_{\text{AR-A}}$, denoted by $\theta_{i,\text{AR-A}}$, with $L_{R1}$ can be written as

$$\dot{\theta}_{i,\text{AR-A}} = \theta_{i,\text{AR-A}} - R\frac{\partial L_2(\text{AR-A}, \alpha_1)}{\partial \theta_{i,\text{AR-A}}}, \tag{9}$$

where $R$ denotes the learning rate, $\dot{\theta}_{i,\text{AR-A}}$ denotes the parameter after backpropagated by gradient decent in a step. $\boldsymbol{\theta}_{\text{AR-B}}$ can be trained by the same mechanism.

If we approximately consider the influence of a training dataset $\alpha_1$ on $\theta_{i,\text{AR-A}}$ as a one step gradient decent (which also means $L_{R1}$ and $L_{R2}$ are optimized at the same time), the influence of a dataset parameter $\theta_{\text{j,AR-D}}$ on $L_{R2}$ through $\theta_{i,\text{AR-A}}$ can then be analyzed. When only considering the parameter $\dot{\theta}_{i,\text{AR-A}}$ in gradient decent, the partial derivative of $L_{R2}$ with the respect to the parameter $\theta_{\text{j,AR-D}}$ in $\alpha_1$ is the multiplication between partial derivative $\dfrac{\partial \dot{\theta}_{i,\text{AR-A}}}{\partial \theta_{\text{j,AR-D}}}$ and $\dfrac{\partial L_{R2}}{\partial \dot{\theta}_{i,\text{AR-A}}}$:

$$\frac{\partial \dot{\theta}_{i,\text{AR-A}}}{\partial \theta_{\text{j,AR-D}}}\frac{\partial L_{R2}}{\partial \dot{\theta}_{i,\text{AR-A}}} = -R\frac{\partial L_{R2}}{\partial \dot{\theta}_{i,\text{AR-A}}}\frac{\partial^2 L_2(\text{AR-A}, \alpha_1)}{\partial \theta_{\text{j,AR-D}}\partial \theta_{i,\text{AR-A}}} = -R\text{Partial}(\text{AR-A}, i, j), \tag{10}$$

where we let the backpropagation of $\theta_{i,\text{AR-A}}$ prior to the backpropagation of $\theta_{j,\text{AR-D}}$ at each step. $\dfrac{\partial^2 L_2(\text{AR-A}, \alpha_1)}{\partial \theta_{\text{j,AR-D}}\partial \theta_{i,\text{AR-A}}}$ can be calculated since $\theta_{\text{j,AR-D}}$ influence $\alpha_1$ in $L_2$.

Then training $\theta_{j,\text{AR-D}}$ when considering the influence of both $\alpha_2$ and $\alpha_1$ can be denoted by

$$\dot{\theta}_{j,\text{AR-D}} = \theta_{j,\text{AR-D}} - R\frac{\partial L_{R2}(\alpha_2(\boldsymbol{\theta}_{\text{AR-D}}))}{\partial \theta_{j,\text{AR-D}}} + R^2\sum_i \text{Partial}(\text{AR-A}, i, j) + R^2\sum_k \text{Partial}(\text{AR-B}, k, j), \tag{11}$$

where $\dfrac{\partial L_{R2}(\alpha_2(\boldsymbol{\theta}_{\text{AR-D}}))}{\partial \theta_{j,\text{AR-D}}}$ denotes the derivative only considering the direct influence of $\theta_{j,\text{AR-D}}$ on testing dataset $\alpha_2$. Since the learning rate $R$ is a small value such as 0.01, the gradient $\text{Partial}(\text{AR-A}, i, j)$ and $\text{Partial}(\text{AR-B}, k, j)$ have small influence on the gradient decent of $\theta_{j,\text{AR-D}}$, which is omitted in our implementation. The gradient decent of other searched dataset variables have the same characteristics. As a result, we only consider the first aspect of the influence on $L_{R2}$ when searching probable datasets with $L_{R2}$.

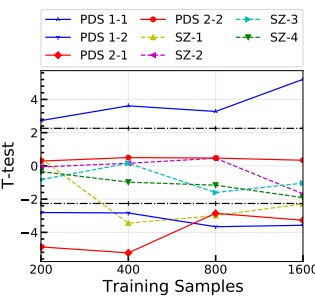
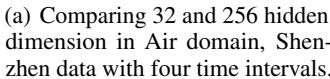
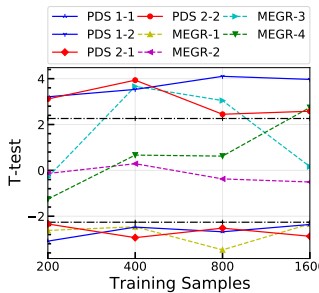
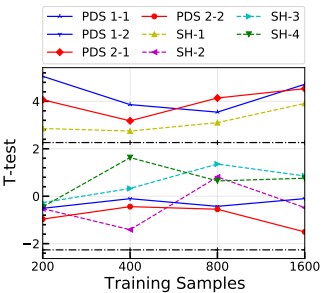

(a) Comparing 32 and 256 hidden dimension in Air domain, Shenzhen data with four time intervals.

(b) Comparing 32 and 256 hidden dimension in HandMEG domain, left hand and wrist movement data (MEGR, MEG Right) with four time intervals.

(c) Comparing 4 and 32 hidden dimension in Air domain, Shanghai data with four time intervals.

Figure 2: Hyper parameter comparisons in real world complete datasets and the probable datasets searched by PDS 1 and 2. The comparisons between hyper parameters are measured by $t$-test score with positive/negative symbol. The black dash-and-dot lines refer to the $t$-test score that has statistically significant difference.

## A.2 SEARCHED PROBABLE DATASETS ILLUSTRATION

In this part, we show the difference between hyper parameters through the $t$-test score, where a $t$-test larger 2.262 means two hyper parameters are statistically significantly different with $5\%$ significance level. For easier representation, when comparing the error value of two architectures named AR-EA and AR-EB, we apply a positive (or negative) symbol on the $t$-test score, which corresponds to the situation that the error value of AR-EA is larger (or smaller) than AR-EB. In the figure, when comparing $n_1$ and $n_2$ hidden dimension, AR-EA and AR-EB correspond to the architectures with $n_1$ and $n_2$ dimension in showing $t$-test score with symbol. Here we show the hyper parameters difference of real world complete datasets and searched probable datasets in Fig. 2, where the $t$-test score with symbol are reported with real word datasets and the datasets searched by PDS 1 and 2 methods in each sub-figure. For example, in Fig. 2(a), SZ-1, 2, 3, 4 refers to the Shenzhen data with forecasting intervals 1, 2, 3, 4. PDS methods are followed with two numbers, where the first number refers to the type of PDS method and the second number corresponds to the architecture whose performance is aimed to be improved in PDS method (second number 1 or 2 corresponds to AR-EA or AR-EB). For example, PDS 2-1 refers to the probable datasets searched in loss Eq. (6) with PDS 2 methods (because of the first number), where the loss aims to search for the probable dataset that AR-EA (corresponds to 32 dimension, because of the second number) performs better, so AR-A and AR-B in loss Eq. (6) corresponds to the LSTM with 32 and 256 hidden dimensions. Similarly, PDS 1-2, PDS 2-2 refer to the probable datasets searched in loss Eq. (6) with PDS 1 and 2 method, where the loss aims to search the probable datasets that 256 dimension performs better. In the figure, if a value is larger or smaller than two black dash-and-dot lines, corresponding comparison has statistical significance.

The results show that with more information required (PDS 1 < PDS 2), PDS method could make better predictions. For example, when comparing hyper parameters between 32 and 256 hidden dimensions in SZ datasets, PDS 1 method could find the probable dataset that 32 or 256 hidden dimension performs significantly better than another, so it makes type 3 predictions. PDS 2 could not find the situation that 256 hidden dimension performs significantly better, so it makes type 1 prediction. Considering the performance of two hyper parameters that 256 hidden dimension does not performs significantly better in any SZ real world dataset, type 1 is a better prediction. When comparing hyper parameter between 4 and 32 hidden dimensions in SH datasets, PDS 1 and 2 both make type 2 predictions although they require different levels of dataset information.

The result in Fig. 2(b) explains the situation that SDP 1 and 2 make wrong predictions, where for the same $X$, there are real world datasets with different $Y$ that 32 or 256 hidden dimension performs significantly better than another. In these datasets, PDS methods make type 3 predictions, which avoid making wrong predictions.

