# OpenReview forum: "Probable Dataset Searching Method with Uncertain Dataset Information in Adjusting Architecture Hyper Parameter"
_ICLR.cc/2023/Conference — Submitted to ICLR 2023_

### Official Review · Reviewer_Au4D · 2022-10-22

**Confidence:** 3
**Correctness:** 4
**Technical Novelty And Significance:** 2
**Empirical Novelty And Significance:** Not applicable
**Recommendation:** 3

**Clarity, Quality, Novelty And Reproducibility:**

Clarity:
The paper is very difficult to follow. The current presentation makes it really hard to assess the paper properly. The reviewer encourages the authors to rewrite the paper focusing on presentation clarity. Below are some suggestions.

•	Abstract, it is unclear what the term “probable dataset”. Please clarify.

•	Introduction. Introduction should contain at least clear answers to the following questions: (1) What the authors try to achieve? (2) Why the authors try to achieve it? (3) How the authors try to achieve it? (4) What are the contributions of the paper? (5) What is the impact of the paper on the research community?

•	Positioning to previously published methods. The introduction does not position the work well w.r.t. previous art. Please position the proposed method w.r.t. previous art it the introduction.

•	Figure 1. Caption should be auto-explanatory. Please extend the text of the caption to simplify the understanding of the figure.

•	Evaluation metric. It is unclear why the AES metric is the proper metric to compare models.

•	Evaluation datasets. The chosen evaluation datasets are rather not commonly used in the ML community. Could the authors comment on why these datasets have been chosen?

•	Evaluation. The model is only compared to two baselines CP and SPD. How would the model compare previously published methods for hyperparameter model prediction (e.g., the ones mentioned in the related work section)?

•	Table 1. Please add standard deviation estimates over multiple random seeds.

•	What are the limitations of the proposed pipeline?


Quality and Novelty:
The quality and novelty of the paper are hard to assess due to presentation clarity. The experimental validation looks rather poor – no comparison to previous methods, the evaluation is performed on rather simple datasets. The limitations of the proposed approach are not discussed. The model uses simple and standard machine learning techniques, this the individual blocks does not have novelty; however, the overall pipeline could have some aspects of novelty that are obscured with current presentation.

Reproducibility:
There is no mentioning about code release. The pipeline components look rather simple and thus the whole pipeline should be reproducible.


**Strength And Weaknesses:**

Strengths:
The idea of predicting model hyperparameters is interesting

Weaknesses (for details, see section below):
The paper is very difficult to follow
The validation is limited
Lack of comparisons to previous works.
Limitations are not discussed


**Summary Of The Paper:**

The paper proposes approach that enables simple hyperparameter search in scenarios where either label information is delayed or when the training on full dataset is too costly.  The paper proses to use an auxiliary neural network to predict missing labels and a method to search for probable dataset to select a subset of data to use for training. The proposed solutions are tested on three datasets (Air, WormMotion, HandMEG) and compared to two baselines – the results show that the introduced methods outperform baselines.

**Summary Of The Review:**

Overall, the paper studies an interesting problem of model hyperparameter prediction on uncertain data. However, the paper have several important weaknesses that makes it hard to recommend the paper for acceptance: (1) the presentation of the paper is very difficult to follow, (2) the evaluation is rather weak, (3) the paper is not properly positioned w.r.t. previous art.

---

### Official Review · Reviewer_xHNU · 2022-10-24

**Confidence:** 4
**Correctness:** 3
**Technical Novelty And Significance:** 2
**Empirical Novelty And Significance:** 2
**Recommendation:** 3

**Clarity, Quality, Novelty And Reproducibility:**

Clarity: The method part is clear while I find it kind of diffucult to follow the experimental section.

Quality: This paper contributes some new ideas but the empirical evaluation is limited.

Novelty: The proposed idea is somehow interesting.

Reproducibility: With the current version, I might feel it's not easy to reproduce the main resutls of this paper as some of the details are missing.

**Strength And Weaknesses:**

**Strengths**:

1. This paper proposes an interesting task: to adjust hyper parameters under different uncertain dataset information settings.

2. The paper proposes several losses to search for probable dataset when the complete dataset information is missing.

3. The paper conducts experiments on multiple real world datasets.

**Weaknesses**:

1. It’s not very clear to me how to train the AR-D. The paper mentions that ‘we let the dataset regression function architecture AR-D be the same with AR-A in $L_{R2}$ since we want to search for the dataset that AR-A performs well and a dataset with architecture AR-A should best fit the AR-A model.’ Initializing AR-D with the parameters of pertain AR-A? Otherwise, if only using the same architecture, is there a guarantee that the AR-A will perform better than AR-B?

2. The experimental part is limited. In the current version, only classification tasks on small datasets have been conducted. I would expect more strong experimental results to support the main claim of the paper. For example, how does the proposed probable dataset searching strategy affect more complex tasks with much bigger datasets?

3. The AES evaluation metric (Eq.8) is not well justified/motivated. How to determine the weights of ‘Wrong’ and ‘Training’?

Minors:

1. I would suggest remaking figure.1 to improve the figure quality.

2. Eq. equation 1 -> Eq.1.

3. A few typos:

  * Page 6, paragraph CP, ‘always make type 3 prediction’ -> ‘always makes type 3 prediction’.

  * Page 6, paragraph SDP 1, ‘will also performs well’ -> ‘will also perform well’.

  * Page 6, paragraph SDP 1, try to be consistent to use dataset 1 or data 1.


**Summary Of The Paper:**

This paper tries to adjust hyper parameters in two types of uncertain dataset information: 1) dataset labels are postponed to be obtained so hyper parameters need to be adjusted without complete dataset information. 2) hyper parameters are adjusted with a subset training dataset since training models with complete training dataset is time consuming. More specifically, this paper proposes several loss functions to search for probable dataset when the complete dataset information is not obtained. The authors conduct experiments on 9 real world datasets to demonstrate the performance of the proposed method.

**Summary Of The Review:**

Overall, this paper contributes some new ideas but the empirical evaluation is limited. Also, the presentation needs to be improved.

---

### Official Review · Reviewer_d4iE · 2022-10-25

**Confidence:** 4
**Correctness:** 2
**Technical Novelty And Significance:** 3
**Empirical Novelty And Significance:** Not applicable
**Recommendation:** 3

**Clarity, Quality, Novelty And Reproducibility:**

The manuscript is written in a fairly clear fashion. The major concern is that the methodological contribution is weak.

**Strength And Weaknesses:**

Strength:
(1) To deal with the uncertain information in a dataset, this paper proposes a probable dataset searching method to predict architecture comparison.
(2) Different experimental settings are studied, and the authors validate the performance of the proposed method in 9 real-world data with two types of uncertain dataset information.

Weakness:
(1) This paper utilizes a neural network to approximate the dataset regression function and apply several loss functions to search for the probable dataset. In this case, the authors adopt several existing technologies, so the technical contribution is weak.
(2) The authors state that they propose a probable dataset searching method to predict architecture comparison. However, many SOTA methods are missing in the comparison experiments.


**Summary Of The Paper:**

This paper mainly discusses adjusting hyper parameters in two types of uncertain dataset information, including the postponed dataset labels and only using a subset training dataset. To infer the probable complete dataset information, this paper presents a probable dataset searching method that searches the variables in our dataset representation. Experimental results have been conducted on 9 real-world data with two types of uncertain dataset information.

**Summary Of The Review:**

This paper mainly discusses adjusting hyper parameters in two types of uncertain dataset information, including the postponed dataset labels and only using a subset training dataset. However, the overall technical novelty should be strengthed and more related SOTA comparisons are needed.

---

### Decision · Program_Chairs · 2023-01-20

**Decision:**

Reject

**Justification For Why Not Higher Score:**

Presentation and empirical comparison should be improved for future submissions.

**Justification For Why Not Lower Score:**

N/A

**Metareview: Summary, Strengths And Weaknesses:**

This paper addresses adjusting hyperparameters in two types of incomplete datasets, including postponed labeling and only using a subset of training dataset. A probable dataset searching method is presented to handle these interesting cases. Dealing with the incomplete dataset is an interesting problem and the approach seems to be sound. However, there are some concerns that should be carefully considered for future submissions. First of all, the presentation should be improved, since reviewers criticized it is not easy to follow to properly asses the work. The empirical comparisons are weak, which should be also improved. Therefore, the paper is not recommended for acceptance in its current form. I hope authors found the review comments informative and can improve their paper by addressing these carefully in future submissions.